:PLOS | ONE

# Circulating CTRP9 correlates with the prevention of aortic calcification in renal allograft recipients

**Nobuhiko Miyatake, Hiroki Adachi, Kanae Nomura-Nakayama, Keiichiro Okada, Kazuaki Okino, Norifumi Hayashi, Keiji Fujimoto, Kengo Furuichi, Hitoshi Yokoyama\*** 

Department of Nephrology, Kanazawa Medical University School of Medicine, Daigaku, Uchinada, Ishikawa, Japan

\* h-yoko@kanazawa-med.ac.jp

## Abstract

### Background

Cardiovascular disease (CVD) due to atherosclerosis is a major cause of death in renal allograft recipients. Recently, C1q/TNF-α related protein-9 (CTRP9), which is a paralog of adiponectin (ADPN), has been suggested to be related to the prevention of atherosclerosis and the occurrence of CVD, but this relationship has not been confirmed in renal allograft recipients.

### Subjects and methods

The relationships among the serum CTRP9 concentration, serum ADPN concentration, and vascular calcification were investigated in 50 kidney transplantation recipients at our hospital. Calcification of the abdominal aorta was evaluated according to the aortic calcification area index (ACAI) calculated from CT images. Changes in the serum CTRP9 and ADPN fractions and ACAI were examined for 8 years. In addition, the expression of CTRP9 and ADPN and their respective receptors AdipoR1 and R2 in muscular arteries of the kidney was examined by immunofluorescence.

### Results

In renal allograft recipients, the serum CTRP9 concentration at the start of the observation was not significant correlated with eGFR or serum high-molecular-weight (HMW)-ADPN concentration (rS = -0.009, p = 0.950; rS = -0.226, p = 0.114, respectively). However, the change in the serum CTRP9 concentration was positively correlated with the change in the serum HMW-ADPN concentration (rS = 0.315, p = 0.026) and negatively correlated with the change in ACAI (rS = -0.367, p = 0.009). Multiple regression analysis revealed that the serum HMW-ADPN concentration was a significant positive factor for the change in the serum CTRP9 concentration. Moreover, for ACAI, an increase in the serum CTRP9 concentration was an improving factor, but aging was an exacerbating factor. Furthermore, colocalization of CTRP9 and AdipoR1 was noted in the luminal side of intra-renal arterial intima.

**Data Availability Statement:** All relevant data are within the manuscript and its Supporting Information files.

**Funding:** This study was supported in part by Grants-in-Aid for Scientific Research from the Japan Society for the Promotion of Science; (C) 18K08256 (HY); Grants for Intractable Renal Disease Research and for Health and Labour Sciences Research from the Ministry of Health, Labour, and Welfare of Japan (HY); a Grant-in-Aid for Investigating New Evidences to Understand the safety of Kidney Transplantation from marginal Donors, Promotion of renal Disease Control Grants from Japan Agency for Medical Research and Development (K Furuichi, HY), Grants for Study at Kanazawa Medical University (No. S2017-7) (HA) and Grants for a Cooperative Study at Kanazawa Medical University (No. C2017-4) (HY).

**Competing interests:** The authors have declared that no competing interests exist.

**Abbreviations:** ACAI, aortic calcification area index; Adipo or ADPN, adiponectin; CTRP9, C1q/TNF-α related protein-9; CVD, cardiovascular disease; eGFR, estimated glomerular filtration rate; hr, hour; sCr, serum creatinine.

## Conclusion

In renal allograft recipients, both CTRP9 and HMW-ADPN were suggested to prevent the progression of aortic calcification through AdipoR1.

## Introduction

Infection, malignant disease, and cardiovascular disease (CVD) are among the major causes of death in renal allograft recipients [1]. CVD is strongly related to atherosclerotic lesions, and the risk of cardiovascular events in renal allograft recipients is reportedly 50-times higher than that in healthy individuals [2]. Vascular calcification, particularly calcification of the coronary artery, is a strong factor related to such cardiovascular events and cardiovascular death. Furthermore, the progression of atherosclerotic lesions leads to ischemic damage of the renal graft and may cause chronic allograft nephropathy, dyslipidemia, and hypertension [3]. Therefore, control of vascular calcification is considered an important issue in both the survival and prognosis of kidney allograft recipients.

Adiponectin (ADPN), which is secreted by fat cells of white and brown adipose tissues, is attracting attention as a factor closely associated with the prevention of coronary artery disease and improvement of insulin sensitivity. ADPN is a physiologically active substance that is secreted by adipose tissue, and acts on local and distant organs. ADPN improves insulin sensitivity, and exhibits anti-diabetic, anti-atherosclerotic, and anti-inflammatory actions, with high-molecular-weight dodecamer and octadecamer ADPN being more closely involved in these actions [4,5]. There are also C1q/TNF-α related protein (CTRP) family proteins as ADPN paralogs that belong to the C1q/TNF protein superfamily along with ADPN. Among CTRP1 to 15, CTRP9 has the most similar structure to ADPN [6]. Specifically, of the 4 dominants that constitute CTRP family proteins, approximately 51% of the amino acids that form the globular C1q domain of CTRP9 are the same as those that constitute the globular domain of ADPN [7]. CTRP9 and ADPN form heterotrimers, share adiponectin receptor 1 (AdipoR1), and are physiologically active on vascular endothelial cells and myocardial cells [8]. Thus, CTRP9 suppresses TNF-α reactive inflammatory reactions by activating AdipoR1-dependent AMP-activated protein kinase (AMPK), and reduces tissue damage caused by oxidation action associated with glucose uptake. However, the blood level of CTRP9 was reported to be reduced in obese model mice, and its characteristics are similar to those of ADPN. The blood CTRP9 concentration was also reduced in a mouse cardiac ischemia-reperfusion model and myocardial infarction model [9]. Thus, in patients with or at a high risk of CVD, CTRP9 secretion may be reduced and its vascular protecting action therefore attenuated.

There has been no report on the relationship of CTRP9 with atherosclerotic lesions in renal allograft recipients. Regarding the relationships of ADPN with lipids, vascular calcification, and cardiovascular complications in patients with chronic kidney graft malfunction, we previously reported that HMW-ADPN is a factor that affects the accumulation of visceral fat, vascular calcification, and the development of CVD [10,11]. In this study, we noted CTRP9, which is a paralog of ADPN, and evaluated its effects on atherosclerosis and CVD in renal allograft recipients by a retrospective cohort study.

## Subjects and methods

### Subjects

Our subjects comprised 50 patients (33 males and 17 females) who had undergone renal transplantation at Kanazawa Medical University Hospital and had serum creatinine levels of ≤3

mg/dL, and in whom the transplant had engrafted by the start of the study period in 2008 and engraftment persisted until 2018. We used serum samples taken at fasting condition in this study, and we had a proper diet guidance and food intake was stable at the time of registration in all patients. We prohibited smoking in renal allograft recipients, and excluded a few current smokers even after transplantation.

The following factors were investigated: the age at transplant, sex, donor type (living or deceased renal donor), the time since the transplant, the estimated glomerular filtration rate (eGFR) at the start of the study period, body mass index (BMI), the serum levels of triglyceride (TG), total cholesterol (T chol), high-density lipoprotein cholesterol (HDL-C), low-density lipoprotein cholesterol (LDL-C), non-HDL-C (= T chol—HDL-C), and each ADPN fraction (high molecular weight (HMW)-, middle molecular weight (MMW)- and low molecular weight (LMW)-ADPN fractions), CTRP-9, immunosuppressive drug use, anti-diabetic drug (including insulin) use, statin use, anti-hypertensive drug use, and bisphosphonate use.

We evaluated 1) the serum levels of lipid markers, CTRP-9 and ADPN fractions, the renal function of the transplanted organs, and the aortic calcification area index (ACAI); 2) the correlations between serum levels of CTRP-9 and ADPN fractions, and vascular calcification; and 3) the correlations between the serum level of CTRP-9 and HMW-ADPN fraction or lipid markers and allograft function (estimated glomerular filtration rate, eGFR) at the start (2008) and the end (2016) of the study period.

Our study did not include any vulnerable populations, such as prisoners, subjects with reduced mental capacity due to illness or age, or children. In addition, we only used blood samples, radiological scans, and renal biopsy specimens in this study. The study protocol was approved by the ethics committee of Kanazawa Medical University (Kanazawa Medical University Epidemiological Study Review No. I127). All patients provided written informed consent, and the study was conducted according to the principles of the Declaration of Helsinki and Istanbul.

## Methods

Serum creatinine levels were analyzed using an enzymatic method, a Hitachi creatinine auto-analyzer (model 7170; Hitachi, Tokyo, Japan), and an enzyme solution (Preauto-SCrE-N; Daiichi Pure Chemicals Co., Tokyo, Japan). The serum levels of T chol, LDL-C, and HDL-C were measured by direct enzymatic assays using an automatic analyzer (Hitachi, Tokyo, Japan). The serum levels of the total, HMW-, MMW-, LMW-ADPN, and CTRP-9 fractions were measured using a sensitive enzyme-linked immunosorbent assay kit (SEKISUI MEDICAL Co., Tokyo, Japan). Renal function was evaluated based on the eGFR (= $194 \times SCr^{-1.094} \times age^{-0.287} \times 0.739$ for females, ml/min/1.73 m$^2$), which was calculated based on the serum creatinine (SCr) level, as described previously [12].

We evaluated calcification of the abdominal aorta using the ACAI. The ACAI was calculated based on assessments of computed tomography scans of the abdominal aortic wall (slice thickness: 5 mm or 10 mm) in the region of interest. Specifically, it was calculated by assessing the percentage of the aortic wall occupied by calcification on each slice and then dividing the sum of the percentage values for all slices by the number of slices (S1 Fig). These analyses were conducted using image analysis software (MITANI Co., Ltd., Fukui, Japan).

**Immuno-histopathological examinations.** Fresh tissue specimens, which were embedded in OCT compound and frozen in acetone-dry ice mixture, were cut at a thickness of 3 μm on a cryostat. The frozen sections were fixed in a 1:1 mixture of acetone and methanol, and then blocked with 10% goat serum in 0.01 mol/L phosphate-buffered saline. Staining of CTRP9, AdipoR1, and AdipoR2 was performed by indirect immunofluorescence using the

primary monoclonal or polyclonal antibodies listed in S1 Table. Anti-mouse or rabbit IgG polyclonal goat IgG antibodies conjugated with Alexa Fluor 488 (Thermo Fisher Scientific, A-11029) or Alexa Flour 555 (Thermo Fisher Scientific, A-21429) were used as secondary antibodies, and their signals were visualized using a BX51/DP71 fluorescence microscope/CCD camera (Olympus IMS, Japan).

**Statistical analysis.** All continuous variables are presented as the median and interquartile range (IQR). The Mann-Whitney test was used for comparisons between sexes and of the change in the ACAI. The relationships between the serum levels of lipid markers and the serum levels of CTRP9 or each ADPN fraction were evaluated using Spearman's correlation coefficient. The factors influencing the serum CTRP9 level, HMW-ADPN level, and the ACAI were analyzed using multivariate regression analysis. Stat Flex version 6 (Artech Co., Ltd., Osaka, Japan) was used as the statistical analysis software.

The formula for Changes in CTRP9, HMW-ADPN and ACAI is shown below.

$$Change\ in\ CTRP9 = \{(serum\ CTRP9\ level\ in\ 2008) - (serum\ CTRP9\ level\ in\ 2016)\}/8$$

$$Change\ in\ HMW\ ADPN = \{(serum\ HMW - ADPN\ level\ in\ 2008) - (serum\ HMW - ADPN\ level\ in\ 2016)\}/8$$

$$Change\ in\ ACAI = \{(ACAI\ in\ 2008) - (ACAI\ in\ 2016)\}/8$$

## Results

### Clinical background

The laboratory test results at baseline are shown in Table 1. Comparisons between sexes demonstrated no significant difference in the duration of dialysis or period after transplantation. Moreover, no significant difference was noted in the eGFR at baseline or serum CTRP9 levels in 2008 or in its subsequent changes. The LDL-C level was significantly higher in males (p = 0.001). The HDL-C level was significantly higher (p = 0.001), and the non-HDL-C level was significantly lower (p = 0.001) in females. The serum HMW-ADPN and MMW-ADPN concentrations were higher in females (p = 0.006, p = 0.005). No significant difference was noted in the history of medication.

### Relationships between the eGFR and serum CTRP9 or ADPN levels in renal graft recipients

The relationships between renal allograft function and the serum levels of CTRP9 and each ADPN fraction are shown in S2 Fig. The CTRP9 and MMW-ADPN level was not significantly correlated with the eGFR (rS = -0.009, p = 0.950, n = 50, rS = -0.218, p = 0.128, n = 50, respectively), whereas the HMW-and LMW-ADPN levels were inversely correlated with the eGFR (rS = -0.319, p = 0.024, n = 50 and rS = -0.372, p = 0.008, n = 50, respectively), as previously reported.

### Relationships between the serum CTRP9 level and the serum levels of each lipid marker

The associations between the serum CTRP9 level and the serum levels of lipid markers at the start of the study period are shown in S2 Table. The CTRP9 level was not correlated with the serum levels of HDL-C, LDL-C, TG, or non-HDL-C (rS = 0.020, p = 0.893; rS = 0.239, p = 0.095; rS = 0.041, p = 0.778 and rS = 0.146, p = 0.312, respectively). The CTRP9 level was

**Table 1. The baseline characteristics.**

| Valuable | total | Male | Female |
|---|---|---|---|
| Donor (living : deceased) | 50 (43:7) | 33 (29:4) | 17 (14:3) |
| Age at transplant (years) | 31.5 (24.0–36.0) | 33.0 (23.5–36.3) | 30.0 (24.3–35.0) |
| Duration of dialysis (years) | 17.9 (8.3–50.2) | 14.7 (8.3–52.6) | 21.2 (8.2–61.3) |
| Time since Tx (months) | 284 (217–367) | 293 (236–370) | 231 (201–330) |
| BMI (kg/m$^2$) | 21.2 (18.8–23.0) | 21.3 (18.9–23.2) | 20.8 (18.8–22.6) |
| eGFR in 2008 (mL/min/1.73m$^2$) | 51.3 (42.2–57.1) | 52.0 (43.1–62.8) | 48.3 (39.3–56.3) |
| eGFR in 2016 (mL/min/1.73m$^2$) | 47.6 (39.4–58.0) | 52.5 (41.1–59.5) | 42.8 (38.1–55.4) |
| ΔeGFR (mL/min/1.73m$^2$/year) | -1.8 (-6.7–2.9) | -0.25 (-0.58–0.41) | -0.2 (-0.99–0.23) |
| Serum Ca (mg/dL) | 9.60 (9.30–9.80) | 9.60 (9.40–9.90) | 9.50 (9.20–9.63) |
| Serum phosphorus (mg/dL) | 2.95 (2.60–3.20) | 2.80 (2.40–3.13) | 3.10 (2.90–3.40) |
| LDL-C (mg/dL) | 103.5 (85.0–123.0) | 113.0 (97.0–132.5)** | 90.0 (76.5–98.3)** |
| HDL-C (mg/dL) | 61.0 (55.0–79.0) | 57.0 (51.0–67.0)** | 78.0 (66.8–96.5)** |
| TG (mg/dL) | 126.0 (92.0–175.0) | 127.0 (108.8–186.3) | 116.0 (85.0–169.0) |
| non-HDL-C (mg/dL) | 129.6 (110.4–153.8) | 144.8 (121.4–167.7)* | 111.4 (97.7–131.5) * |
| Blood glucose (mg/dL) | 95.5 (87.0–109.0) | 97.0 (87.8–122.8) | 92.0 (81.0–99.0) |
| Total ADPN in 2008 (µg/mL) | 7.44 (5.39–10.20) | 6.65 (4.96–8.43)* | 10.20 (7.43–11.36)* |
| HMW-ADPN (µg/mL) | 2.71 (1.68–4.52) | 2.48 (1.62–3.33)** | 4.52 (3.02–6.79)** |
| MMW-ADPN (µg/mL) | 1.76 (1.18–2.26) | 1.67 (1.14–1.89)** | 2.26 (1.85–2.83)** |
| LMW-ADPN (µg/mL) | 2.62 (2.14–3.33) | 2.49 (2.15–3.34) | 2.89 (2.01–3.34) |
| CTRP9 in 2008 (ng/mL) | 2.05 (2.00–2.13) | 2.08 (2.01–2.13) | 2.03 (2.00–2.07) |
| ACAI in 2008 | 0.10 (0.00–1.30) | 1.72 (0.00–1.55) | 0.00 (0.00–1.26) |
| Therapeutic agents (drug use, %) | | | |
| Steroids | 50 (100%) | 33 (100%) | 17 (100%) |
| Anti-metabolites | 48 (96%) | 32 (96%) | 16 (94%) |
| Calcineurin inhibitors | 37 (74%) | 24 (73%) | 13 (76%) |
| Anti-hypertensive drugs | 40 (80%) | 29 (88%) | 11 (65%) |
| Anti-diabetic drugs | 4 (8%) | 3 (9%) | 1 (6%) |
| Insulin | 2 (4%) | 2 (6%) | 0 (0%) |
| Oral anti-diabetic drugs | 2 (4%) | 1 (3%) | 1 (6%) |
| Statins | 35 (70%) | 19 (58%) | 16 (94%) |
| Bisphosphonates | 9 (18%) | 5 (15%) | 4 (24%) |

Abbreviations: Tx, transplant; eGFR: estimated glomerular filtration rate, LDL-C: low-density lipoprotein cholesterol, HDL-C: high-density-lipoprotein cholesterol, TG: triglyceride, ADPN: Adiponectin, HMW: high-molecular-weight, MMW: middle- molecular-weight, LMW: low-molecular-weight, ACAI: aortic calcification area index, CTRP9: C1q/TNF-α related protein-9.

Data are shown as median (IQR) values.

*:<0.05

**:<0.01, male vs. female according to the Mann-Whitney test.

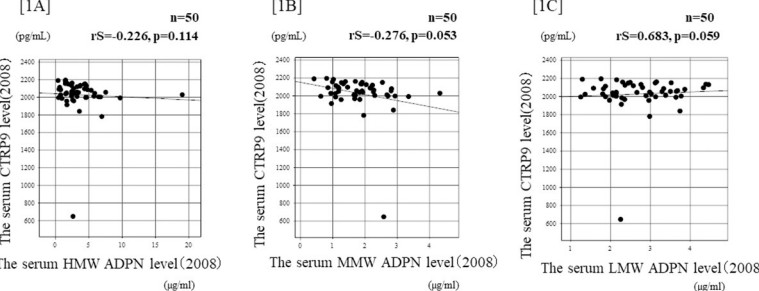

**Fig 1. Relationships between the serum CTRP9 and ADPN levels in renal allograft recipients.** [1A] The serum CTRP concentration at baseline was not significantly correlated with the serum HMW-ADPN concentration in the same year (rS = -0.226, p = 0.114, according to Spearman's rank coefficient). [1B] The serum CTRP concentration at baseline was not significantly correlated with the serum MMW-ADPN concentration in the same year (rS = 0.129, p = 0.053, according to Spearman's rank coefficient). [1C] The serum CTRP concentration at baseline was not significantly correlated with the LMW-ADPN concentration in the same year (rS = 0.683, p = 0.059, according to Spearman's rank coefficient).

not correlated with BMI or the change in BMI (rS = 0.138, p = 0.340 and rS = 0.021, p = 0.884, respectively).

## Relationships between the changes in CTRP9 and ADPN

The relationships between the serum CTRP9 and ADPN concentrations at baseline are shown in Fig 1. In 2008, the serum CTRP concentration was not correlated with the serum HMW-, MMW-, or LMW-ADPN concentration in the same year ([1A] rS = -0.226, p = 0.114, [1B] rS = 0.129, p = 0.053, and [1C] rS = 0.683, p = 0.059). The relationship between the change in the serum CTRP9 concentration and the change in the serum ADPN concentration is shown in Fig 2. The change in the serum HMW-ADPN concentration was positively correlated with the change in the serum CTRP9 concentration ([2A rS = 0.437, p = 0.015]. However, the change in the serum MMW- or LMW-ADPN concentration was not significantly correlated with the change in the serum CTRP9 concentration ([2B] sR = 0.129, p = 0.371 and [2C] rS = 0.087, p = 0.549). In addition, no significant change of adiponectin fractions during study period over at most 8 years in most cases.

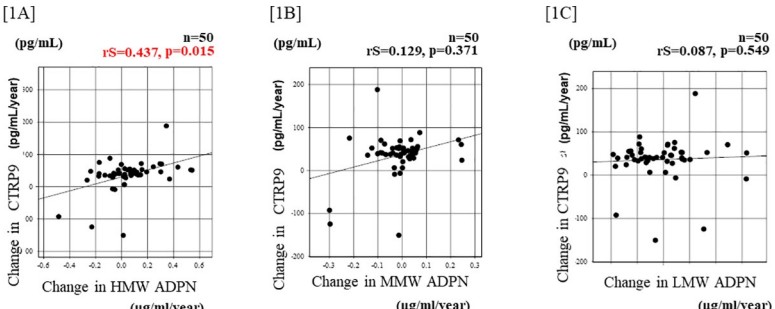

**Fig 2. Relationships between the change in ADPN and CTRP9 in renal allograft recipients.** [2A] The change in the serum HMW-ADPN concentration during 8 years was positively correlated with the change in the serum CTRP9 concentration during the same period (rS = 0.437, p = 0.015, according to Spearman's rank coefficient). [2B] The change in the serum MMW-ADPN concentration was not significantly correlated with the change in the serum CTRP9 concentration during the same period (rS = 0.129, p = 0.371, according to Spearman's rank coefficient). [2C] The change in the serum LMW-ADPN concentration was not significantly correlated with the change in the serum CTRP9 concentration (rS = 0.087, p = 0.549, according to Spearman's rank coefficient).

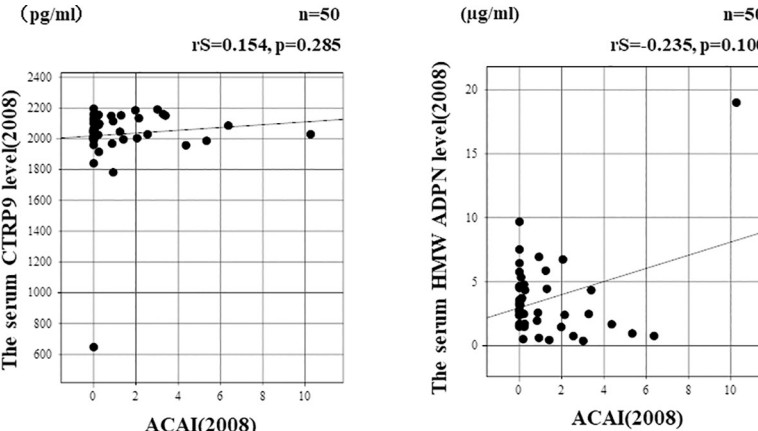

**Fig 3. Relationships between the serum CTRP9 level and ACAI in renal allograft recipients.** [3A] The serum CTRP concentration at baseline was not significantly correlated with ACAI in the same year (rS = 0.154, p = 0.285, according to Spearman's rank coefficient). [3B] The serum HMW-ADPN concentration at baseline was not significantly correlated with ACAI in the same year (rS = -0.235, p = 0.100, according to Spearman's rank coefficient).

## Relationships between the change in ACAI and CTRP9

The relationships of the serum CTRP9 and HMW-ADPN concentrations at baseline with ACAI are shown in Fig 3. The serum CTRP9 or HMW-ADPN concentration in 2008 was not correlated with ACAI in the same year ([3A] rS = 0.154, p = 0.285 and [3B] rS = -235, p = 0.100, respectively). The relationships of the change in ACAI with the changes in the serum CTRP9 and ADPN concentrations after 8 years are shown in Fig 4. The change in the serum CTRP9 concentration was negatively correlated with the change in ACAI ([4A] rS = -0.367, p = 0.009]. The change in the serum HMW-ADPN concentration was also negatively correlated with the change in ACAI ([4B] rS = -0.542, p = 0.004).

## The factors influencing the change in CTRP9 and ACAI in renal transplant subjects

The factors that affected the change in the serum CTRP9 concentration are presented in Table 2. In the multiple regression analysis using the change in the serum CTRP9

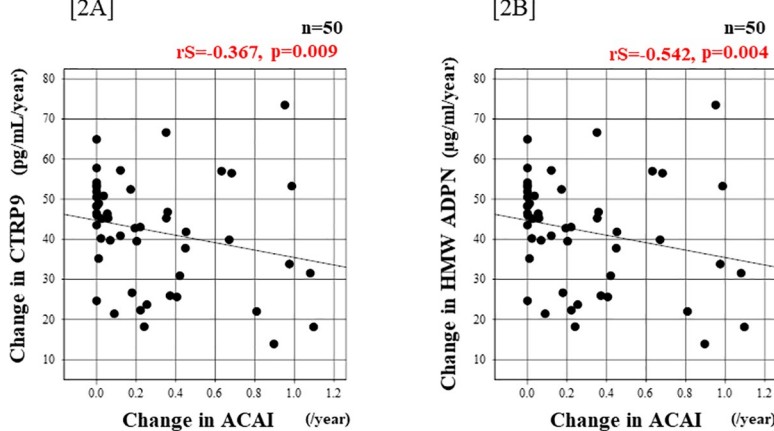

**Fig 4. Relationship between the change in ACAI and CTRP9 in renal allograft recipients.** [4A] The change in the serum CTRP9 concentration during 8 years was negatively correlated with the change in ACAI (rS = -0.367, p = 0.009, according to Spearman's rank coefficient). [4B] The change in the serum HMW-ADPN concentration was also negatively correlated with the change in ACAI (rS = -0.542, p = 0.004, according to Spearman's rank coefficient).

**Table 2. Factors influencing the change in CTRP9 in renal transplant subjects.**

| Objective variable: Change in CTRP9 | | | | | |
|---|---|---|---|---|---|
| | β | SE | std | p | t score |
| (Constant) | 135.3 | 80.50 | | | |
| Age at transplantation (years) | -0.648 | 0.822 | -0.116 | 0.435 | 0.788 |
| Sex (Male = 1, Female = 2) | -9.206 | 16.95 | -0.089 | 0.590 | 0.543 |
| BMI (kg/m$^2$) | -1.207 | 2.313 | 0.076 | 0.605 | 0.522 |
| HDL-C (mg/dL) | -0.122 | 2.313 | -0.048 | 0.790 | 0.268 |
| Duration after transplantation (years) | -0.133 | 0.084 | -0.216 | 0.124 | 1.571 |
| Change in HMW-ADPN (μg/mL/year) | 97.24 | 34.51 | 0.394 | 0.007 | 2.818 |

CTRP9: C1q/TNF-α related protein-9, HMW-ADPN: high-molecular-weight adiponectin, LMW-ADPN: low-molecular-weight adiponectin, ACAI: aortic calcification area index, HDL-C: high-density lipoprotein cholesterol, BMI: body mass index. Explanatory variables: age at transplantation (year), sex, HDL-C (mg/dL), duration after transplantation (year), and change in HMW-ADPN (μg/mL/year). According to multivariate regression analysis.

concentration as the objective variable, an increase in change in the serum HMW-ADPN concentration was found to be a significant positive factor.

Regarding factors that affected the change in ACAI, old age at transplantation was an exacerbating factor, whereas an increase in change in the serum CTRP9 concentration was an improving factor (Table 3). Similarly, an increase in change in the serum HMW-ADPN concentration was an improving factor of the change in ACAI (Table 4).

## Renal expression of CTRP9, AdipoR1, and AdipoR2

The findings concerning the expression of CTRP9, and colocalization of AdipoR1 and R2 on vascular endothelial cells in fresh frozen sections of renal graft tissues are presented in Fig 5. The expression of CTRP9 and AdipoR1 was confirmed in the luminal side of intima, and AdipoR2 expression was confirmed primarily in the media of the arterial wall. Colocalization of CTRP9 and AdipoR1 was demonstrated by merging images, but none was noted about AdipoR2.

**Table 3. Factors influencing ACAI in renal transplant subjects.**

| Objective variable: Change in ACAIModel 1 | | | | | |
|---|---|---|---|---|---|
| | β | SE | std | p | t score |
| (Constant) | -0.414 | 0.506 | | | |
| Age at transplantation (years) | 0.038 | 0.113 | 0.376 | 0.011 | 2.639 |
| Sex (Male = 1, Female = 2) | -0.043 | 0.111 | 0.054 | 0.740 | 0.334 |
| BMI (kg/m$^2$) | 0.015 | 0.015 | 0.141 | 0.329 | 0.987 |
| HDL-C (mg/dL) | 0.000 | 0.003 | 0.013 | 0.943 | 0.072 |
| Duration after transplantation (year) | 0.000 | 0.001 | 0.040 | 0.773 | 0.291 |
| Change of CTRP9 (pg/mL/year) | -0.002 | 0.001 | 0.288 | 0.041 | 2.110 |

CTRP9: C1q/TNF-α related protein-9, ACAI: aortic calcification area index, HDL-C: high-density lipoprotein cholesterol, BMI: body mass index.

Explanatory variables: age at transplantation (year), sex, BMI, HDL-C (mg/dL)

Duration after transplantation (year), and change in CTRP9 (pg/mL/year).

According to multivariate regression analysis.

**Table 4. Factors influencing ACAI in renal transplant subjects.**

| Objective variable: Change in ACAIModel 2 | | | | | |
|---|---|---|---|---|---|
| | β | SE | std | p | t score |
| (Constant) | -0.448 | 0.476 | | | |
| Age at transplantation (years) | 0.013 | 0.005 | 0.347 | 0.017 | 2.491 |
| Sex (Male = 1, Female = 2) | -0.036 | 0.110 | -0.051 | 0.741 | 0.332 |
| BMI (kg/m$^2$) | 0.014 | 0.014 | 0.130 | 0.352 | 0.941 |
| HDL-C (mg/dL) | 0.000 | 0.003 | -0.003 | 0.985 | 0.019 |
| Duration after transplantation (year) | 0.000 | 0.001 | 0.097 | 0.463 | 0.740 |
| Change in HMW-ADPN (µg /mL/year) | -0.610 | 0.223 | -0.362 | 0.009 | 2.733 |

Explanatory variables: age at transplantation (year), sex, BMI, HDL-C (mg/dL)

Duration after transplantation (year), and change in HMW-ADPN (µg/mL/year)

According to multivariate regression analysis.

## Discussion

This study provided three important results. First, the change in the serum CTRP9 concentration was closely associated with the change in the serum HMW-ADPN concentration in renal

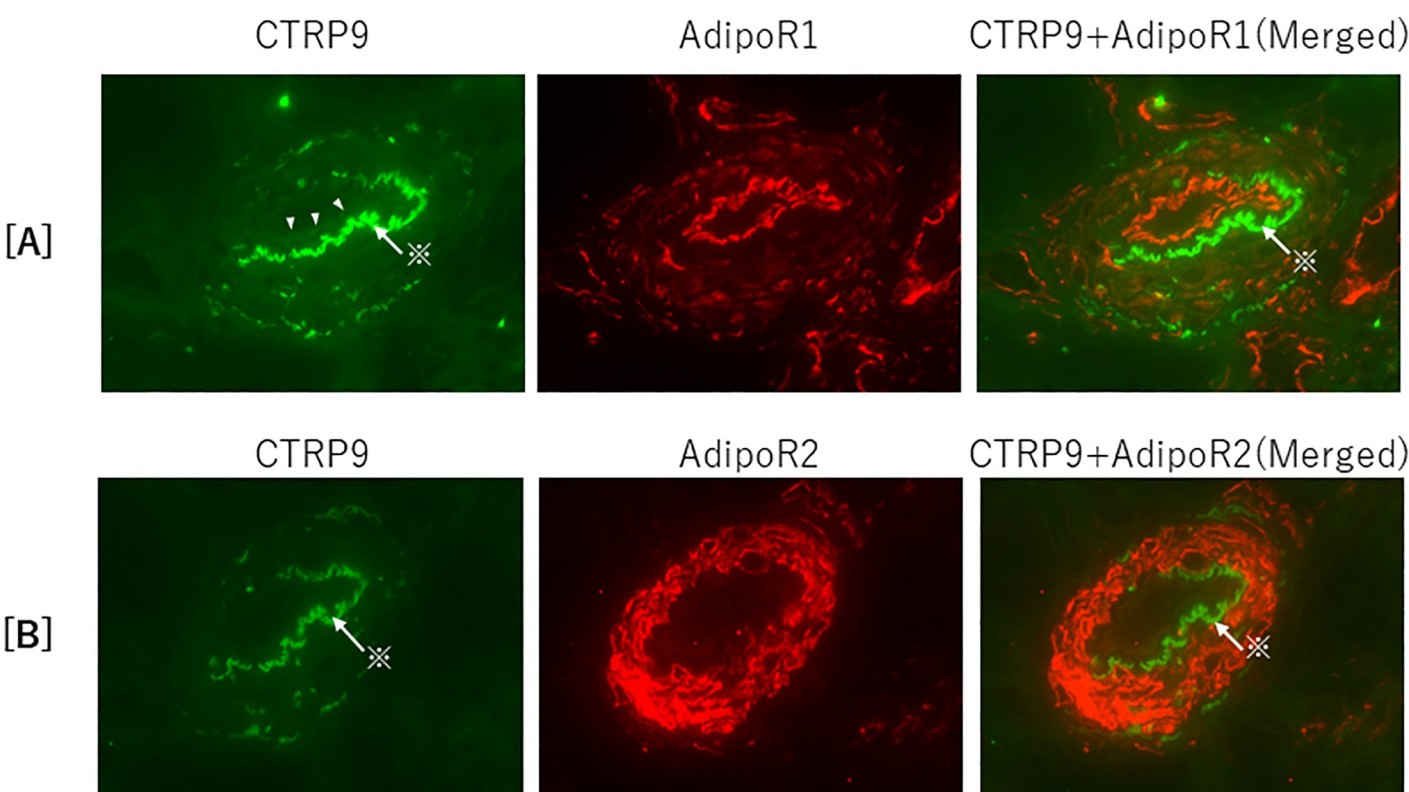

**Fig 5. Renal expression of CTRP9 and AdipoR1/R2 in renal allografts.** Renal biopsy specimens were examined by immunofluorescent multi-staining using anti-CTRP9, anti-AdipoR1, and anti-AdipoR2 antibodies. By merging the images, colocalization of CTRP9 and AdipoR1 was observed in the vascular intima [5A], but not in the media of AdipoR2 [5B].

allograft recipients. Second, the change in the serum CTRP9 concentration was negatively correlated with the change in ACAI, suggesting that CTRP9 can be a suppressive factor for vascular calcification. Third, the expression of CTRP9 and colocalization of CTRP9 and AdipoR1 were confirmed by fluorescent immunostaining of renal biopsy specimens.

In this study, no significant correlation was observed among the serum CTRP9 and ADPN concentrations at baseline and ACAI. Therefore, their relationship was evaluated using their annual changes. As a result, the change in the serum CTRP9 concentration also exhibited a significant positive correlation with the change in the serum HMW-ADPN concentration in renal allograft recipients. When the multiple regression analysis was performed using the change in the serum CTRP9 concentration as the objective variable, an increase in the serum HMW-ADPN concentration was a significant increasing factor. Concerning this, a relationship between the serum CTRP9 and ADPN concentrations was reported in a previous study using mice. For example, in an obese mouse model, the serum CTRP9 concentration was reduced to a similar degree as the serum ADPN concentration [8]. First, CTRP9 and ADPN are both adipocytokines produced by fat cells, and as they are produced by the same tissue, they have close structural similarity. In addition, their expression has been suggested to be suppressed in inflamed tissues. For example, the expression of ADPN is suppressed when insulin resistance-inducing factors, such as TNFα, increase [13]. Furthermore, CTRP9 production was also reduced significantly in mice in which inflammation was induced in adipose tissue [14]. Therefore, if the same tissue develops chronic inflammation and tissue damage, the production of both CTRP9 and ADPN may be suppressed, possibly explaining the correlation of their changes.

We previously reported that HMW-ADPN is a factor significantly related to the accumulation of visceral fat, vascular calcification, and development of CVD in renal allograft recipients. Moreover, in this study, the serum CTRP9 concentration is also a vascular protective factor in renal allograft recipients. Incidentally, ADPN is a factor that is influenced by several lipid markers, obesity, and the accumulation of visceral fat. For example, it has been demonstrated that the serum ADPN concentration is significantly reduced in obese patients, that the serum HMW-ADPN concentration is negatively correlated with the visceral fat area and eGFR [10], and that the serum HMW-ADPN and non-HDL-C concentrations are negatively correlated [11]. In this study, however, the serum CTRP9 concentration at baseline was not correlated with any of the lipids examined. Moreover, the visceral fat area is known to be closely correlated with BMI [15]. However, no correlation was observed between the serum CTRP9 concentration and eGFR or BMI, or between the serial changes in the serum CTRP9 concentration and BMI in this study. This suggests that CTRP9 acts as an independent factor unaffected by the renal graft function, lipids, or visceral fat, unlike ADPN.

Next, we directed our attention to calcification of the abdominal aorta, which is measured as ACAI, and assessed its changes and relationship with CTRP9. In patients with chronic kidney diseases, vascular calcification is classified into atherosclerotic intimal calcification, which is observed primarily in the aorta and carotid artery, and Möncheberg's medial calcification, which is observed primarily in peripheral middle-sized arteries and arterioles. The two types are mixed in the coronary artery [16]. Atherosclerotic intimal calcification and Möncheberg's medial calcification have been demonstrated to be exacerbating factors for all-cause mortality, cardiovascular mortality, and cardiovascular morbidity in the general population [17]. In addition, in patients with end-stage renal failure, calcification of the abdominal aorta is a significant prognostic factor for all-cause and cardiovascular deaths [18], and its progression was evaluated according to ACAI in this study. As a result, serial changes in ACAI in renal allograft recipients exhibited negative correlations with changes in the serum CTRP9 and HMW-ADPN concentrations. Factors for vascular calcification and atherosclerosis include

death of vascular smooth muscle cells and their differentiation into osteogenic/chondrogenic cells, degeneration/degradation of elastin, and remodeling of the vascular wall [19]. Degenerated lipids, such as oxidized LDL, generated under oxidative stress, inflammatory cytokines (TNF-α, IFN-γ, IL-1β), and NO are also involved in smooth muscle cell death [20]. In Möncheberg's medial calcification, stimuli, including inflammation of the medial smooth muscle layer in the arterial wall, oxidative/mechanical stress, and AGE (advanced glycation end-product), induce vascular wall remodeling and calcification [21]. As CTRP9 and ADPN improve insulin sensitivity and have anti-diabetic and anti-inflammatory activity, they may exert suppressive effects on the above mechanism and slow the progression of vascular calcification. Indeed, in patients with coronary artery disease and type 2 diabetes, the serum CTRP9 concentration was found to be correlated with the accumulation of cell adhesion molecules in the vascular wall [22]. In addition, a study using human vascular tissues suggested that CTRP9 activates AMPK via AdipoR1 and R2 receptors in common with ADPN in venous smooth muscle cells [23]. Moreover, there is a cAMP/PKA-dependent pathway and AMPK-dependent pathway for the development of the function of CTRP9 [24, 25]. Through this pathway, CTRP9 prevents intimal thickening and vascular remodeling in mechanically injured blood vessels [26]. Calcification of the abdominal aorta may have been suppressed because of such physiological actions of CTRP9. Furthermore, calcification of the coronary artery has been reported to be observed more frequently in patients with insulin resistance than in the control group, suggesting a similar disease state [27]. A decrease in the serum HMW-ADPN concentration is also a risk factor for an increase in cardiovascular events in patients with atherosclerosis, hypertension, and dyslipidemia [28]. As increases in CTRP9 and HMW-ADPN were ACAI-improving factors in this study, CTRP9 may have prevented vascular calcification.

In this study, we examined the localization of CTRP9 and its colocalization with AdipoR1 and R2 in the muscular artery wall. CTRP9 was colocalized with AdipoR1 but not with AdipoR2. Regarding the relationship between CTRP9 and AdipoR1, a recent study in mice suggested that CTRP9 suppresses nerve cell apoptosis after cerebral hemorrhage through the AdipoR1-dependent pathway [29]. Similarly, the possibility that high glucose-induced vascular endothelial injury is prevented through the AdipoR1-dependent pathway has also been reported [30]. In a study using fish, the expression of AdipoR1 mRNA was inhibited, but AdipoR2 was not, after the administration of globular CTRP9 (gCTRP9), suggesting a close relationship between CTRP9 and AdipoR1 [31]. In this study, the expression of CTRP9 was consistent with that of AdipoR1 in the intima of human intrarenal muscular arteries; therefore, CTRP9 function was suggested to be mediated by AdipoR1 *in vivo*.

### Limitations

Limitations of this study include 1) the small number of cases, 2) retrospective nature of the study, 3) not using high accuracy inflammatory markers such as sensitive CRP or TNF-α, and 4) limited amount of stained renal biopsy specimens. Given the small number of subjects, is the study at risk for bias due to the population studied (Japanese population) and finding may not be more generalized to other populations. In the future, it will be necessary to evaluate vascular lesions, renal function, inflammation markers, and prognosis by prospective studies, and advances in the histological evaluation of CTRP9 expression in the vascular endothelium, atherosclerosis, and vascular calcification are needed.

### Conclusion

In renal allograft recipients, the serum CTRP9 concentration was suggested to change, being positively correlated with the high-molecular-weight ADPN concentration without being

affected by the renal function, to prevent the progression of aortic calcification. In addition, AdipoR1 was also suggested to be involved in the function of CTRP9 at the intima of human arteries.

## Supporting information

**S1 Fig. The evaluation method of abdominal aortic calcification.** It was calculated by assessing the percentage of the aortic wall occupied by calcification on each slice and then dividing the sum of the percentage values for all slices by the number of slices.
(TIF)

**S2 Fig. The relationships between renal allograft function and the serum levels of CTRP9 and each ADPN fraction.** The CTRP9 and MMW-ADPN level was not significantly correlated with the eGFR (rS = -0.009, p = 0.950, n = 50, rS = -0.218, p = 0.128, n = 50, respectively), whereas the HMW-and LMW-ADPN levels were inversely correlated with the eGFR (rS = -0.319, p = 0.024, n = 50 and rS = -0.372, p = 0.008, n = 50, respectively), as previously reported.
(TIF)

**S1 Table. The species, dilution values, and sources of the primary and secondary antibodies.**
(TIF)

**S2 Table. Correlations between the serum CTRP9 level and the concentration of each lipid marker.**
(TIF)

**S3 Table. CTRP9 dataset.**
(XLSX)

## Acknowledgments

The authors gratefully acknowledge the help and assistance of their colleagues at the Department of Nephrology.

## Author Contributions

**Conceptualization:** Hiroki Adachi, Hitoshi Yokoyama.

**Data curation:** Nobuhiko Miyatake, Hiroki Adachi, Kanae Nomura-Nakayama.

**Formal analysis:** Nobuhiko Miyatake, Hiroki Adachi.

**Funding acquisition:** Hitoshi Yokoyama.

**Investigation:** Nobuhiko Miyatake.

**Project administration:** Hiroki Adachi.

**Supervision:** Hiroki Adachi, Kanae Nomura-Nakayama, Keiichiro Okada, Kazuaki Okino, Norifumi Hayashi, Keiji Fujimoto, Kengo Furuichi, Hitoshi Yokoyama.

**Writing – original draft:** Nobuhiko Miyatake.

**Writing – review & editing:** Hitoshi Yokoyama.

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
