## [Decision Letter · Decision Letter 0]

3 Sep 2019

PONE-D-19-20735

Circulating CTRP9 correlates with the prevention of aortic calcification in renal allograft recipients

PLOS ONE

Dear Dr. Yokoyama,

Thank you for submitting your manuscript to PLOS ONE. After careful consideration, we feel that it has merit but does not fully meet PLOS ONE’s publication criteria as it currently stands. Therefore, we invite you to submit a revised version of the manuscript that addresses the points raised during the review process.

Sorry for delay of review processes.  Two experts raised several serious concerns to draw your conclusion.  Please revise the manuscript following their comments.

We would appreciate receiving your revised manuscript by Oct 18 2019 11:59PM. To enhance the reproducibility of your results, we recommend that if applicable you deposit your laboratory protocols in protocols.io, where a protocol can be assigned its own identifier (DOI) such that it can be cited independently in the future. For instructions see: http://journals.plos.org/plosone/s/submission-guidelines#loc-laboratory-protocols

We look forward to receiving your revised manuscript.

Kind regards,

Tatsuo Shimosawa, M.D., Ph.D.

Academic Editor

PLOS ONE

Journal Requirements:

2. We noticed you have some minor occurrence(s) of overlapping text with the following previous publication(s), which needs to be addressed:

https://doi.org/10.1371/journal.pone.0195066

In your revision ensure you cite all your sources (including your own works), and quote or rephrase any duplicated text outside the Methods section. Further consideration is dependent on these concerns being addressed.

Additional Editor Comments (if provided):

Reviewers' comments:

Reviewer's Responses to Questions

**Comments to the Author**

1. Is the manuscript technically sound, and do the data support the conclusions?

Reviewer #1: Partly

Reviewer #2: Partly

2. Has the statistical analysis been performed appropriately and rigorously? 

Reviewer #1: Yes

Reviewer #2: Yes

3. Have the authors made all data underlying the findings in their manuscript fully available?

Reviewer #1: Yes

Reviewer #2: Yes

4. Is the manuscript presented in an intelligible fashion and written in standard English?

Reviewer #1: Yes

Reviewer #2: Yes

5. Review Comments to the Author

Reviewer #1: This manuscript entitled “Circulating CTRP9 correlates with the prevention of aortic calcification in renal allograft recipients” written by Miyatake et al showed that the change in the serum CTRP9 concentration was positively correlated with the change in the serum HMW-ADPN concentration, and that multiple regression analysis showed that the change in the serum CTRP9 concentration was negatively associated with the change in ACAI. These findings are potentially interesting and useful data to understand the increased development of atherosclerotic lesions in renal allograft recipients. However, their data do not have the important confounding factors, such as high sensitive CRP or TNF-α. In addition, immuno-histological staining of Adipo R1 and R2 is quite poor images.

1. As the authors mentioned in Discussion, the concentration of CTRP9 and HMW-ADPN is strongly associated with the inflammatory state. Thus, the authors should measure some inflammatory mediators like high sensitive CRP, TNF-α, IL-6.

2. In Figure 5, the staining of Adipo R1 and R2 looks non-specific. These data do not show the proof of co-localization between CTRP9 and Adipo R1. The authors should exhibit more clear stain. Or, Figure 5 should be removed.

Reviewer #2: Novelty:

The manuscript is the 1st report on the negative correlation of circulation CTRP9 and aortic calcification in renal transplantation patients.

Major issues:

1. Renal allograft IS (co-location of CTRP9 and AdipoR1) was not able to lead the conclusion ‘CTRP9 and HMW-ADPN were suggested to prevent the progression of aortic calcification through AdipoR1.’ Because chronic rejection implicates the allograft vessels directly. To address this concern, animal allogeneic solid organ transplantation model (e.g. heart, kidney or liver) could be set and non-relevant artery (abdominal artery) should be stained for co-location to exclude the direct allorejection background.

2. What’s the relationship of concentration of CTRP9 (and HMW-ADPN) with ACAI at endpoint (2016)?

3. How were the “concentration changes” calculated? This should be described in “Method” section.

6. PLOS authors have the option to publish the peer review history of their article (what does this mean?). If published, this will include your full peer review and any attached files.

Reviewer #1: No

Reviewer #2: No

---

## [Author Response · Author response to Decision Letter 0]

12 Nov 2019

Dear to Editor,

Thank you very much for your thoughtful comments.

We would like to re–submit the revised manuscript entitled “Circulating CTRP9 correlates with the prevention of aortic calcification in renal allograft recipients’’.

We are most grateful to you and the reviewers for the helpful comments on the original version of our manuscript. 

We have taken all these comments into account and submit.

According to your reviewer’s comments, we improved our manuscripts as below.

The reviewer-1 comments:

This manuscript entitled　“Circulating CTRP9 correlates 

with the prevention of aortic calcification in renal allograft recipients” written by Miyatake et al showed that the change in the serum CTRP9 concentration was positively correlated with the change in the serum HMW-ADPN concentration, and that multiple regression analysis showed that the change in the serum CTRP9 concentration was negatively associated with the change in ACAI. These findings are potentially interesting and useful data to understand the increased development of atherosclerotic　 lesions in renal allograft recipients. However, their data do not have the important confounding factors, such as high sensitive CRP or TNF��. In addition, immuno-histological staining of Adipo R1 and R2 is quite poor images.

Comment #1:

As the authors mentioned in Discussion, the concentration of CTRP9 and HMW-ADPN is strongly associated with the inflammatory state. Thus, the authors should measure some inflammatory mediators like high-sensitive CRP, TNF��, IL-6.

Response:

We think it difficult to measure the mediators such as high-sensitivity CRP, TNF-α, and IL-6, as you pointed out. One reason is that we thought they were not significantly different because we measured them in a stable state, and another reason is the limited amount of sample needed for measurement.

Comment #2:

In Figure 5, the staining of Adipo R1 and R2 looks non-specific. These data do not show the proof of co-localization between CTRP9 and Adipo R1. The authors should exhibit more clear stain. Or, Figure 5 should be removed.

Response:

We re-stained in another case for Adipo R1 and R2 IS (immunofluorescent staining), to confirm more specific staining. Hence, the conditions of reagents used for staining have been changed. As a result, more specific staining was obtained for CTRP9 and AdipoR1. A clearer colocalization was confirmed. We can send the newly obtained image in a manuscript here.

The reviewer-2 comments:

The manuscript is the 1st report on the negative correlation of circulation CTRP9 and aortic calcification in renal transplantation patients.

Comment #1:

Renal allograft IS (co-location of CTRP9 and AdipoR1) was not able 

to lead the conclusion ‘CTRP9 and HMW-ADPN were suggested to prevent the progression of aortic calcification through AdipoR1. Because chronic rejection implicates the allograft vessels directly. To address this concern, animal allogeneic solid organ transplantation model (e.g. heart, kidney or liver) could be set and non-relevant artery (abdominal artery) should be stained for co-location to exclude the direct allorejection background.

Response:

In this study, the proof of CTRP9 and AdipoR1 collocation in renal allograft IS in this study was aimed to show that the CTRP9 expression site coincides with the localization of AdipoR1. As you pointed out, we think that this IS alone cannot prove the process of suppressing vascular calcification. In addition, animal experiments of organ transplantation models, such as your proposal, will be difficult to carry out at our facility.

Comment #2:

What’s the relationship of concentration of CTRP9 (and HMW-ADPN) 

with ACAI at endpoint (2016)?

Response:

We couldn’t confirm significant correlation CTRP9 and HMW-ADPN concentrations with concurrent ACAI at the 2016 endpoint.

Comment #3: 

How were the “concentration changes” calculated? This should be described in “Method” section. 

Response:

We used the calculation method of concentration change is shown below.

Change in CTRP9= {(serum CTRP9 level in 2008)- (serum CTRP9 level in 2016)}/8

Change in HMW ADPN= {(serum HMW-ADPN level in 2008)- (serum HMW-ADPN level in 2016)}/8

Change in ACAI= {(ACAI in 2008)- (ACAIin 2016)}/8

The method for calculating the concentration change is shown in the Method section.

We will continue to make corrections centered on the above points and send the revised manuscript by the due date.

We thank for your advices.

Nobuhiko Miyatake, MD C/O Hitoshi Yokoyama, MD, PhD

Kanazawa Medical University School of Medicine

Department of Nephrology

1-1 Daigaku, Uchinada, Ishikawa 920-0293, JAPAN.

Tel: +81-76-218-8166

FAX: +81-76-286-2786

E-mail: nobuhiko@kanazawa-med.ac.jp

---

## [Decision Letter · Decision Letter 1]

21 Nov 2019

PONE-D-19-20735R1

Circulating CTRP9 correlates with the prevention of aortic calcification in renal allograft recipients

PLOS ONE

Dear Dr. Yokoyama,

Thank you for submitting your manuscript to PLOS ONE. After careful consideration, we feel that it has merit but does not fully meet PLOS ONE’s publication criteria as it currently stands. Therefore, we invite you to submit a revised version of the manuscript that addresses the points raised during the review process.

The authors could not measure inflammatory markers because they did not plan it and store samples.  It is a limitations of this study.  Authors stated that under stable condition, those markers does not differ.  If so, authors should show scientific proof.

We would appreciate receiving your revised manuscript by Jan 05 2020 11:59PM. To enhance the reproducibility of your results, we recommend that if applicable you deposit your laboratory protocols in protocols.io, where a protocol can be assigned its own identifier (DOI) such that it can be cited independently in the future. For instructions see: http://journals.plos.org/plosone/s/submission-guidelines#loc-laboratory-protocols

We look forward to receiving your revised manuscript.

Kind regards,

Tatsuo Shimosawa, M.D., Ph.D.

Academic Editor

PLOS ONE

Reviewers' comments:

Reviewer's Responses to Questions

**Comments to the Author**

1. If the authors have adequately addressed your comments raised in a previous round of review and you feel that this manuscript is now acceptable for publication, you may indicate that here to bypass the “Comments to the Author” section, enter your conflict of interest statement in the “Confidential to Editor” section, and submit your "Accept" recommendation.

Reviewer #1: (No Response)

Reviewer #2: All comments have been addressed

2. Is the manuscript technically sound, and do the data support the conclusions?

Reviewer #1: Partly

Reviewer #2: Yes

3. Has the statistical analysis been performed appropriately and rigorously? 

Reviewer #1: Yes

Reviewer #2: Yes

4. Have the authors made all data underlying the findings in their manuscript fully available?

Reviewer #1: Yes

Reviewer #2: Yes

5. Is the manuscript presented in an intelligible fashion and written in standard English?

Reviewer #1: (No Response)

Reviewer #2: Yes

6. Review Comments to the Author

Reviewer #1: In my previous comment in the authors’ first submission, inflammatory mediators like hs-CRP, TNF-α or IL6 should be measured because the concentration of CTRP9 and HMW-ADPN is strongly associated with the inflammatory state.

However, he authors has not measured at all.

In addition, re-staining of Figure 5 has not successfully shown the colocalization between CTRP9 and AdipoR1.

In total, the authors has not appropriately revised the manuscript at all.

Reviewer #2: Most of my comments were well addressed, although some of the suggested experimental methods was not able to performed technically.

In the revision, the fonts in line 183-188 are significantly larger than others. Please format.

Furthermore, some of the Figure legends were inserted into the ‘Result’ section, which confused me.

The manuscript is suitable for publication if the above issues are corrected.

7. PLOS authors have the option to publish the peer review history of their article (what does this mean?). If published, this will include your full peer review and any attached files.

Reviewer #1: No

Reviewer #2: No

---

## [Author Response · Author response to Decision Letter 1]

28 Nov 2019

Date: November 26, 2019

Editor office “PLOS ONE”

Dear to Editor,

Thank you very much for your thoughtful comments.

We would like to re–submit the revised manuscript entitled “Circulating CTRP9 correlates with the prevention of aortic calcification in renal allograft recipients’’.

We are most grateful to you and the reviewers for the helpful comments on the original version of our manuscript. 

We have taken all these comments into account and submit.

According to your reviewer’s comments, we improved our manuscripts as below.

Comment #1:

Thank you very much for your thoughtful comment.

Unfortunately, this study was a retrospective study, then the collected samples were insufficient in some cases. Therefore, it was difficult to measure additionally and to compare with other suggested molecules such as high sensitivity CRP and TNF-α. We added the following sentence in limitation; 3) not using high accurate inflammatory markers such as sensitive CRP or TNF��.

However, the concentration of CTRP9 or HMW-ADPN is also strongly associated with the inflammatory state as we mentioned in Discussion.

We hope our revision certified your comment. 

We thank you again for your advice.

Nobuhiko Miyatake, MD C/O Hitoshi Yokoyama, MD, PhD

Kanazawa Medical University School of Medicine

Department of Nephrology

1-1 Daigaku, Uchinada, Ishikawa 920-0293, JAPAN.

Tel: +81-76-218-8166

FAX: +81-76-286-2786

E-mail: nobuhiko@kanazawa-med.ac.jp

---

## [Editor Report · Decision Letter 2]

3 Dec 2019

Circulating CTRP9 correlates with the prevention of aortic calcification in renal allograft recipients

PONE-D-19-20735R2

Dear Dr. Yokoyama,

We are pleased to inform you that your manuscript has been judged scientifically suitable for publication and will be formally accepted for publication once it complies with all outstanding technical requirements.

With kind regards,

Tatsuo Shimosawa, M.D., Ph.D.

Academic Editor

PLOS ONE
---

## [Editor Report · Acceptance letter]

10 Jan 2020

PONE-D-19-20735R2 

Circulating CTRP9 correlates with the prevention of aortic calcification in renal allograft recipients 

Dear Dr. Yokoyama:

I am pleased to inform you that your manuscript has been deemed suitable for publication in PLOS ONE. Congratulations! Your manuscript is now with our production department. 

With kind regards,

on behalf of

Prof. Tatsuo Shimosawa 

Academic Editor

PLOS ONE